# The Role of Peace Attitudes on Sustainable Behaviors: An Exploratory Study

**DOI:** 10.3390/bs14020120

**Published:** 2024-02-07

**Authors:** Rosa Angela Fabio, Alessandra Croce

**Affiliations:** 1Department of Economy, University of Messina, Via dei Verdi, 75, 98122 Messina, Italy; 2Department of Clinical and Experimental Medicine, University of Messina, Via Bivona, 98122 Messina, Italy; alessandracroce@outlook.com

**Keywords:** peace attitude, personality, sustainability, sustainable behaviors

## Abstract

This study delves into the intricate relationship among peace attitudes, personality traits, and sustainable behaviors in a diverse sample of 279 adults from different regions of Italy. Building upon the existing literature, this research affirms the influence of agreeableness, openness, and conscientiousness as primary personality traits associated with sustainable behaviors. Additionally, this study scrutinizes the unique predictive power of peace attitudes. The Peace Attitude Scale (PAS), the Big Five Questionnaire (BFQ), and the Sustainable Behaviors Scale (SBS) were utilized to evaluate peace attitudes, personality traits, and sustainable behaviors. The analysis reveals that peace attitudes significantly predict sustainable behaviors, accounting for 31% of the variance. This predictability is attributed to intrinsic motivation and value alignment. Importantly, peace attitudes extend beyond environmental concerns to embrace social justice and equity, integral components of sustainability. The findings underscore the unique and substantial contribution of peace attitudes to understanding sustainable behavior. This study not only confirms the role of personality traits but also emphasizes the importance of intrinsic values in propelling pro-environmental actions.

## 1. Introduction

The intricate relationship between peace and sustainability, extensively illuminated by Neufeldt et al. [1] and the United Nations Environment Programme (UNEP), extends beyond the realm of conflict. While conflict inflicts severe consequences on the environment, leading to ecosystem devastation and resource depletion, it is crucial to acknowledge that challenges to sustainability persist even in ‘business-as-usual’ conditions of peace. The pursuit of peace remains intimately intertwined with sustainable development, which strives to meet present needs without compromising the welfare of future generations. This interconnectedness contributes to peace by addressing the root causes of conflicts, alleviating poverty, fostering economic opportunities, and preserving natural resources. The concept of human security emphasizes that peace and sustainability are integral components of individual well-being, necessitating fundamental needs to be met within a stable and secure environment [2].

In this context, the present study aims to explore whether peace attitudes, correlated with personality traits, can effectively predict environmentally sustainable behaviors. Recognizing the interdependence between peace, personality, and sustainability, this research seeks to provide a nuanced understanding of the dynamics shaping pro-environmental actions. This study’s specific objectives include analyzing the sub-dimensions of personality to discern their role in sustainable behaviors and investigating the intertwined nature of peace attitudes and sustainable actions. The potential findings hold the promise of informing targeted interventions to promote sustainable attitudes and behaviors while uncovering the alignment between peace attitudes and environmentally friendly actions.

### 1.1. Climate Change and Sustainable Consumption Behaviors

In recent years, the heightened focus on climate change has prompted increased interest in understanding environmental behavior. This behavior encompasses various terms like “pro-environment behavior” [3,4], “sustainable consumption behaviors” [5,6,7,8,9,10], “ecological behaviors” [9,11,12], and “conservation behaviors” [13,14].

Stern’s [15,16] impactful definition of environmental behavior, emphasizing its tangible impact on the environment, extends from public sphere actions, influencing policies, to private sphere behaviors, encompassing individual and household actions. Sustainable consumption, crucial in balancing environmental, social, and economic consequences, involves purchasing eco-friendly products, reducing resource consumption, and prioritizing recycling [17]. In this context, pro-environmental behaviors have been defined as concrete actions, intentional or not, that have a positive impact on the natural environment [18,19].

Building upon the findings of Stern [15,16] and Luchs et al. [17], the research conducted by Fabio, Croce, and Calabrese [20] delves into the broader concept of environmental behavior within sustainability. Expanding investigations to individuals integrating ethical products [21] and understanding mechanisms of consumer behavior [22], their work aims to provide a comprehensive understanding of factors influencing sustainable practices.

In scrutinizing the discourse surrounding climate change and environmental behavior, a spectrum of viewpoints emerges, posing some challenges. For example, Lindzen [23] has expressed skepticism regarding the anthropogenic causes of climate change. On the economic front, Lomborg [24] has argued against an exclusive focus on environmental policies, advocating for the efficient allocation of limited resources to maximize overall social benefits. Stavins [25] has delved into the unequal economic burdens associated with climate policies.

Within the realm of sustainable consumption, Juliet Schor has critically examined the paradox of encouraging eco-friendly purchasing habits potentially contributing to resource depletion and waste [26]. Turning to the realm of political divisiveness, Roger Pielke Jr., a noted political scientist, has extensively studied the politicization of climate issues and its ramifications for environmental policies [27]. Petrou et al.’s work on environmentalism and cultural theory sheds light on the variations in environmental perceptions and practices across different cultures and regions [28]. This multifaceted analysis offers a comprehensive view of the complexities influencing the pursuit of sustainability.

### 1.2. Factors Influencing Environmentally Friendly Behavior

Understanding the motivations behind environmentally friendly behavior is crucial, especially in the face of growing awareness of climate change and the urgent need for concrete actions. Previous research has uncovered various factors associated with ecological attitudes and behaviors, including locus of control, personal responsibility, economic orientation, moral norms, social norms, intention, and feelings of guilt [6,29,30,31,32,33,34]. Personality traits have also been linked to ecological attitudes and behaviors [35]. Studies have indicated that individuals characterized by openness to new experiences and relationships, as well as traits of cooperativeness, generosity, and friendliness, tend to feel a profound connection to nature [5,36,37,38]. Openness stands out as strongly associated with pro-environmental attitudes and behaviors, whereas agreeableness, extraversion, and conscientiousness show comparatively weaker links [7,8,39]. Neuroticism, conversely, appears to lack a significant relationship with pro-environmental behaviors or attitudes [35].

Personality traits and attitudes toward peace have also been examined [40,41,42]. Openness has demonstrated a strong association with intergroup attitudes, such as peace and tolerance for outgroup members [9,11,12]. Additionally, agreeableness and conscientiousness have been identified as contributors to peace attitudes [43]. Individuals with strong peace attitudes exhibit affection, sociability, talkativeness, and passion towards others. They are marked by trust, generosity, leniency, and emotional stability; fostering relationships; and employing empathic skills to understand the perspectives of others.

Recognizing the interplay between peace attitudes and pro-environmental attitudes holds significance across multiple dimensions, including intrinsic motivation, value alignment, a holistic approach, proactive problem-solving orientation, social influence, and considerations of environmental justice and equity. In terms of intrinsic motivation and value alignment, peace attitudes often emanate from intrinsic values such as harmony, cooperation, and empathy, aligning closely with the fundamental principles of sustainability. Individuals exhibiting strong peace attitudes may inherently prioritize environmental well-being, viewing it as an extension of their broader commitment to fostering peaceful coexistence [44]. Regarding its holistic approach, peace attitudes may foster a comprehensive perspective that underscores interconnectedness and interdependence, mirroring the holistic nature of sustainability. Those with a peace-oriented mindset may acknowledge the interrelated dynamics among social, economic, and environmental factors, advocating for a more holistic approach to sustainable behaviors. In the context of a proactive problem-solving orientation, individuals embracing peace attitudes may demonstrate a preference for non-confrontational resolutions and sustainable problem-solving strategies in addressing conflicts. This orientation can manifest in proactive engagement with sustainable behaviors, as individuals with peace attitudes actively seek constructive contributions to environmental challenges. Concerning social influence and cooperation, peace attitudes often involve a preference for cooperation and collaboration over conflict. Individuals strongly aligned with peace attitudes may be more inclined to participate in collective sustainable actions, recognizing the imperative of collaborative efforts in addressing environmental concerns.

In the realm of environmental justice and equity, peace attitudes may extend to a commitment to social justice, integral to sustainability [45,46]. Individuals prioritizing peace may be more predisposed to engage in sustainable behaviors that contribute to environmental justice, acknowledging the interconnectedness of social and environmental issues [47].

### 1.3. Aim of This Study

The present research aims to explore the intricate relationships among peace attitudes, personality traits, and sustainable behaviors. After examining the connections between peace attitudes and personality, as well as personality and sustainability, the primary goal is to develop a deeper understanding of the intricate dynamics linking these three constructs.

This research is grounded in a theoretical perspective, suggesting that peace attitudes, personality traits, and sustainable behaviors are interconnected in a complex manner. The approach is exploratory, seeking to provide a more detailed insight into the factors influencing sustainable behaviors. Acknowledging the relatively limited body of specific research on this relationship, this investigation focuses on a theoretical perspective that recognizes conceptual interconnections. The aim is to contribute to the theoretical understanding of how peace attitudes may align with environmental behaviors.

This research objective, therefore, is to contribute to a theoretical understanding of the intricate web of relationships among peace attitudes, personality, and sustainability.

Additionally, this study aims to outline potential implications for future interventions aimed at promoting sustainable attitudes and behaviors. Building on previous findings, the first hypothesis posits that agreeableness, openness, and conscientiousness are primary personality traits associated with sustainable behaviors, a connection consistent in prior studies. Furthermore, the second aim is to investigate the interconnections between peace attitudes and sustainable behaviors, recognizing their intertwined nature with various influencing factors.

## 2. Method

### 2.1. Participants

Two hundred seventy-nine participants (192 females and 87 males) with an average age of 31.16 years (±11.01) were selected through a convenience sampling method. The sampling process involved recruiting individuals from various regions of Italy via widely used social media platforms such as Instagram and Facebook. In an online survey, explicit informed consent was secured from all participants prior to the administration of self-report questionnaires. Demographic information, including gender distribution and socio-economic conditions, was systematically gathered through self-reported measures. Most of the participants, 158 out of 279 subjects, reported being economically dependent (56.6% of the sample), and 226 out of 279 subjects reported living with someone (81.1% of the sample). Table 1 shows the demographic statistics of the sample of 279 subjects.

### 2.2. Instruments

This study utilized three measurement instruments: the Big Five Questionnaire (BFQ) for assessing personality traits based on the Five-Factor Model, the Peace Attitude Scale (PAS) for gauging attitudes toward peace, and the Sustainable Behaviors Scale (SBS) for evaluating sustainable behaviors.

### 2.3. Peace Attitude Scale

The Peace Attitude Scale (PAS) is a 22-item self-report measure developed by Broccoli et al. [30]. This scale comprises five dimensions: socio-political factor, personal well-being factor, ease with diversity, environmental attitude, and caring factor. Respondents rate each statement on a seven-point Likert-type scale, ranging from “Never” to “Always”. Higher scores on the PAS indicate stronger peace attitudes. For example, an item from the socio-political factor is: “I think people need to dialogue with one another in a harmonious way”. An item from the personal well-being factor is: “When something is wrong, I work hard to relax and get back to a state of well-being”. An example (reversed) item related to the ease with diversity factor is: “I would be afraid if I were living in an Islamic State”. An item reflecting the environmental attitude is: “I’d like to clean dirty public places even if it wasn’t me who soiled them”. An item from the caring factor is: “If I bumped into an injured animal, I wouldn’t hesitate to take care of it”.

The Cronbach’s alpha value for reliability is 0.93, indicating high internal consistency. Criterion validity was assessed by correlating the PAS with Neff’s self-compassion scale, resulting in a correlation coefficient of r = 0.56 (*p* < 0.001).

### 2.4. Big Five Questionnaire

The Big Five Questionnaire (BFQ) consists of 132 statements, validated for Italian speakers, assessing the five personality traits: openness to experience, conscientiousness, extraversion, agreeableness, and neuroticism [48,49]. Participants respond using a five-point Likert scale, ranging from “Disagree Strongly” to “Agree Strongly”. Openness to experience refers to needs for variety, novelty, and change; conscientiousness refers to a strong sense of purpose and high aspiration levels; extraversion refers to a preference for companionship and social stimulation; agreeableness refers to a willingness to defer to others during interpersonal conflict; and neuroticism refers to a tendency to experience dysphoric affect such as sadness, hopelessness, and guilt. An example item from the openness trait is “I’m fascinated by novelties”, an example from conscientiousness is “I always pursue the decisions I’ve made through to the end”, an example of the extraversion trait is “I am an active and vigorous person”, an example of agreeableness is “I hold that there’s something good in everyone”, and an example of neuroticism (reversed) is “Usually I don’t lose my calm”. The Cronbach’s alpha value is 0.88.

### 2.5. Sustainable Behaviors Scale

The Sustainable Behaviors Scale (SBS) assesses sustainable behaviors, adapted for Italy from the theory proposed by Luchs et al. [17] (2011) by Fabio, Croce, and Calabrese [20]. Comprising 16 statements, participants rate their responses on a seven-point Likert scale (1 = Never, 2 = Rarely, 3 = Sometimes, 4 = Half the time, 5 = Often, 6 = Almost always, and 7 = Always). It contains items such as: “I regularly use public transportation, bike, or walk instead of driving alone in a car”, “I avoid buying products that are made with non-recyclable or non-biodegradable materials”, “I try to reduce the amount of energy I use at home (e.g., turning off lights when not needed, using energy-efficient appliances)”. The Cronbach’s alpha value is 0.85.

### 2.6. Data Analysis

IBM SPSS 24.0 was utilized for data analysis. The descriptive statistics of the dependent variables were tabulated and examined. Correlational analysis was performed among PAS subscales, BFQ subscales, and SBS subscales with Bonferroni correction. The Bonferroni correction was applied to mitigate the risk of a type I error. The alpha level of 0.05 was adjusted for multiple comparisons involving the six outcome measures, resulting in an alpha level of 0.008. In the second group of correlations, the alpha level of 0.05 was adjusted for multiple comparisons involving the five outcome measures, resulting in an alpha level of 0.01. For all other statistical tests, the alpha level was set to 0.05. In cases of significant effects, the test’s effect size was reported, and effect sizes were computed and categorized according to Cohen [50]. Please refer to the Appendix A for the dataset.

## 3. Results

Before assessing the results, logistic regression analyses were conducted to investigate the influence of socio-economic variables on measures of peace, sustainability, and personality traits in this research. The findings suggest that the independent variables of economic independence, level of instruction, and living alone did not serve as predictors for the Peace Attitude Scale (PAS), the sustainability scale, and personality traits. The logistic regression analyses yielded a non-significant odds ratio of approximately 0.95 (*p* > 0.06), suggesting that there were no statistically significant associations between them. Controlled for gender, the odds ratio for females compared with males was 1.5 (*p* < 0.01) for PAS and 1.4 (*p* < 0.05) for the sustainability scale, suggesting that, when accounting for gender, females exhibited a statistically significant higher likelihood (1.5 times for PAS and 1.4 times for the sustainability scale) compared to males in endorsing favorable attitudes towards peace and sustainable behaviors. Moreover, applying regression analysis with age as the predictor and utilizing the Peace Attitude Scale (PAS), the sustainability scale, and personality traits as dependent variables did not yield statistically significant results.

The results are presented in the following order: (a) descriptive statistics of the personality traits subscales, peace attitudes subscales, and sustainable behaviors subscales; (b) correlational analysis between personality traits, peace attitudes, and sustainable behaviors; (c) regression analysis. Table 2 shows the descriptive statistics of each measurement: the Big Five Questionnaire (BFQ), Peace Attitude Scale (PAS), and Sustainable Behaviors Scale (SBS).

Table 3 shows the correlations between the Peace Attitude Scale, the Big Five Questionnaire subscales, and the Sustainable Behaviors Scale scores. As shown, there are highly significant positive relationships between PAS total score and SBS score (r = 0.527, *p* < 0.0001), between SBS score and openness (r = 0.455, *p* < 0.0001), and between SBS score and agreeableness (r = 0.433, *p* < 0.0001). Table 4 shows the correlations between the Peace Attitude subscales and Sustainable Behaviors Scale scores. As shown, there are highly significant positive relationships between PAS subscales and SBS scores.

The hypotheses were tested by conducting multiple regression models, one with the five personality traits and one with the five value dimensions of the PAS predicting sustainable behaviors. The literature partially confirms that agreeableness, openness, and conscientiousness are the primary personality traits associated with sustainable behaviors; agreeableness and openness predict the SBS score, while conscientiousness does not. The PAS accounts for more variance in predicting sustainable behaviors than personality traits. Comparing the explained model variances (R2), peace attitudes accounted for 31%, while personality accounted for 28% in PAS scoring. The role of peace attitudes as a good predictor of sustainable behavior compared to personality was similarly confirmed. An exploratory post hoc regression model, including all personality traits and peace dimensions simultaneously, was conducted. The results showed that personality and peace together accounted for more than a third of the variance in sustainable behavior (R^2^ = 0.37). Peace attitude total score (β = 0.29), agreeableness (β = 0.21), and openness (β = 0.22) were the best predictors (Table 5).

## 4. Discussion

This study’s findings are consistent with prior research, indicating that specific personality traits significantly influence sustainable behaviors. As suggested by earlier studies [6,29,30,31,39] the present study confirms that agreeableness and openness are primary predictors of sustainable behaviors. Individuals with high levels of agreeableness tend to be cooperative, empathetic, and accommodating [39], fostering prosocial and environmentally friendly behaviors. Openness, characterized by receptivity to new experiences, also emerged as a robust predictor, aligning with findings that individuals open to new experiences are more likely to connect with nature and exhibit pro-environmental behaviors [36,37,38].

This study extends our understanding by exploring the interplay between peace attitudes and sustainable behaviors. As established in the literature [40,41,42,44] peace attitudes exhibit strong correlations with certain personality traits. Notably, individuals characterized by openness tend to embrace peace attitudes, aligning with the concept of tolerance for outgroup members [9,11,12]. This connection between peace attitudes and personality traits is a compelling finding, emphasizing the multifaceted role of personality in influencing sustainable behaviors.

This finding underscores the robust interconnection between the concepts of peace and sustainability, crucial for global human development [10] (Amadei, 2021; and achieving a better future. Sustainable development contributes to peace by reducing poverty, providing economic opportunities, and conserving natural resources, thereby mitigating conflicts and competition over resources. The results also reveal that peace attitude is correlated with all personality traits, with the strongest association found with openness, neuroticism, and agreeableness. These findings align with previous research indicating that individuals with high peace attitudes possess traits such as sociability, generosity, emotional stability, and a willingness to establish and maintain relationships [40,41,42].

Furthermore, this study indicates that sustainable behaviors are highly associated with openness and agreeableness but not with neuroticism and extraversion. Intriguingly, conscientiousness did not significantly predict sustainable behaviors, contrary to the previous literature [10,51]. Instead, openness and agreeableness emerged as the primary predictors of ecological behaviors, suggesting that a sense of duty may not be the primary driver of sustainable behavior adoption. The specific sub-dimensions of openness (openness to experience and openness to culture) and agreeableness (cordiality and cooperativeness) were highly correlated with the adoption of sustainable behaviors.

Understanding the specific sub-dimensions of openness and agreeableness that are strongly correlated with sustainable behaviors provides valuable insights for targeted interventions. By recognizing the importance of peace attitudes and specific personality traits, interventions and policies can be designed to promote sustainable behaviors effectively. Tailoring interventions to individuals with certain personality profiles, particularly those characterized by high openness and agreeableness, can enhance the success of sustainability initiatives [52,53].

### 4.1. Limitations of This Study

It is important to acknowledge the limitations of this study. One limitation is the relatively small sample size and the non-representativeness of the sample in relation to the overall population. Caution should be exercised in generalizing the findings, and further research with larger, more diverse samples is needed to validate the results. Future studies should also consider conducting similar investigations in different countries to explore potential similarities and differences in the relationships between personality traits, peace attitudes, and sustainable behaviors.

Moreover, the use of a convenience sampling method, while efficient for participant recruitment, may limit the generalizability of this study’s findings. The non-randomized selection process could introduce selection bias, and caution should be exercised when extrapolating the results to broader populations.

Another limitation arises from the unequal distribution of gender within the sample. The sample predominantly consists of female respondents (approximately 70%) compared to male respondents (approximately 30%). This gender imbalance could introduce a potential source of bias in this study’s findings, as attitudes and behaviors might vary between genders. Despite efforts to control for gender in statistical analyses, the results should be interpreted with caution, considering the impact of this limitation on the generalizability of the findings.

Another limitation is that a significant number of participants (approximately 60%) reported being unable to make independent economic decisions, including managing their own expenses. This aspect could impact the feasibility of certain sustainable behaviors, as many consumption decisions are tied to financial resources. We acknowledge that the ability to make economic decisions is a relevant factor in the dynamics of sustainable behavior and may pose a limitation in generalizing the results to contexts with greater economic autonomy. Further investigations should address how limited economic autonomy may influence specific sustainable behaviors and identify effective strategies for promoting them in similar contexts.

Another limitation is the self-selection bias: participants who engage with social media platforms might possess distinctive characteristics compared to those who do not, introducing self-selection bias. This factor should be considered when interpreting the results.

Finally, this study’s exploratory focus, while valuable for generating hypotheses, necessitates further research to confirm and generalize the observed relationships.

### 4.2. Conclusions

The findings underscore the interconnection between peace and personality, highlighting the significant role of openness and agreeableness in promoting sustainable behaviors. The findings of this research offer valuable insights for designing targeted interventions aimed at promoting sustainable attitudes and behaviors. Recognizing the importance of both personality traits and peace attitudes, interventions can be tailored to individuals’ unique dispositions. For instance, promoting open-mindedness and empathy through educational and awareness programs may encourage pro-environmental behaviors. By fostering peace attitudes, interventions can effectively motivate individuals to engage in sustainable practices and take actions to protect the environment.

A notable aspect of this study is the comparison between personality traits and peace attitudes as predictors of sustainable behaviors. The results revealed that peace attitudes explained a greater portion of the variance in sustainable behaviors (31%) compared to personality traits (28%). This underscores the potential impact of peace attitudes on driving pro-environmental actions. Considering both traits together, the model accounted for over a third of the variance in sustainable behaviors (R^2^ = 0.37). This implies that a combination of personality traits, especially agreeableness and openness, along with peace attitudes, provides a comprehensive understanding of the factors influencing sustainable behaviors.

## 5. Future Research and Applications

Future research in this area can explore the nuances of how specific peace attitudes, such as tolerance, empathy, and cooperation, relate to distinct dimensions of sustainable behavior, including sustainable consumption, waste reduction, and environmental activism. Additionally, this study’s implications for interventions can be further investigated to develop evidence-based strategies for promoting sustainable attitudes and actions.

In conclusion, this research reinforces the intricate connections between peace attitudes, personality traits, and sustainable behaviors. It highlights the relevance of these constructs in addressing the pressing need for environmentally sustainable actions, emphasizing that both personality and peace attitudes are integral in shaping pro-environmental behaviors. Understanding these dynamics is crucial for advancing a more harmonious and sustainable future, where individuals are not only attuned to the needs of the environment but also to the principles of peace and cooperation.

## Figures and Tables

**Table 1 behavsci-14-00120-t001:** Sociodemographic characteristics of participants at baseline.

	N	% of Total Sample
Female	192	68.8%
Male	87	31.2%
Economically independent	121	43.4%
Living alone	53	18.9%
Age (M(SD))	31.16 (±11.01)	
University graduates	139	50%
High school graduates	90	32%
Middle school	50	18%

**Table 2 behavsci-14-00120-t002:** Means and standard deviations of Big Five Questionnaire, Peace Attitude Scale, and Sustainable Behaviors Scale.

	Means (Standard Deviation)
Big Five Questionnaire	
Openness	85.90 (±11.18)
Neuroticism	67.91 (±15.92)
Agreeableness	81.56 (±9.84)
Extraversion	75.54 (±11.22)
Conscientiousness	84.22 (±10.97)
Peace Attitude Scale	
PAS Total Scoring	114.30 (±11.99)
Sociopolitical	49.28 (±5.71)
Personal well-being	27.24 (±6.14)
Ease with diversity	14.42 (±3.26)
Environmental attitude	13.71 (±3.69)
Caring	9.62 (±2.05)
Sustainable Behaviors Scale	88.33 (±13.66)

**Table 3 behavsci-14-00120-t003:** Pearson’s correlations between Sustainable Behaviors Scale, Big Five Personality subscales, and Peace Attitude Scale.

	Sustainable Behaviors	Openness	Neuroticism	Agreeableness	Extraversion	Conscientiousness	PAS Tot.
Sustainable Behaviors	-						
Openness	0.455 ***	-					
Neuroticism	0.148	0.103	-				
Agreeableness	0.433 ***	0.598 ***	0.189 **	-			
Extraversion	0.188 *	0.377 **	0.127	0.250 **	-		
Conscientiousness	0.160 *	0.340 **	0.004	0.214 *	0.361 **	-	
PAS Tot.	0.527 ***	0.485 ***	0.366 **	0.463 ***	0.255 **	0.314 **	-

* *p* < 0.05; ** *p* < 0.01; *** *p* < 0.001.

**Table 4 behavsci-14-00120-t004:** Pearson’s correlations between Sustainable Behaviors Scale and Peace Attitude subscales.

	Sustainable Behaviors	PAS Total Scoring	Sociopolitical	Personal Well-Being	Ease with Diversity	Environmental Attitude	Caring
Sustainable Behaviors	-						
PAS Total Scoring	0.527 ***	-					
Sociopolitical	0.611 ***	0.701 ***	-				
Personal well-being	0.131	0.218 **	0.189 *	-			
Ease with diversity	0.348 **	0.477 **	0.127	0.180 *	-		
Environmental attitude	0.670 ***	0.240 **	0.231 *	0.214 *	0..298 **	-	
Caring	0.527 ***	0.385 ***	0.366 **	0.299 *	0.301 **	0.356 **	-

* *p* < 0.05; ** *p* < 0.01; *** *p* < 0.001.

**Table 5 behavsci-14-00120-t005:** Comparing personality traits and Peace Attitude Scale in predicting sustainable behaviors.

	R²	R Change	F	β	t	*p*
Big Five	0.28	0.35	19.23	0.297	3.11	0.001
Openness				0.284	2.479	0.01
Agreeableness				0.22	2.315	0.012
Conscientiousness				0.05	0.99	0.56
Neuroticism				0.01	0.33	0.23
Extraversion				0.023	0.215	0.83
PAS Total Scoring	0.31	0.366	18.359	0.341	2.843	0.006
Sociopolitical				0.241	2.46	0.016
Personal well-being				0.119	1.331	0.072
Ease with diversity				0.221	2.321	0.01
Environmental attitude				0.311	2.438	0.01
Caring				0.331	2.316	0.01

## Data Availability

Data are available on request to each of the authors.

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
