# Peer review of "The Role of Peace Attitudes on Sustainable Behaviors: An Exploratory Study"

_behavsci, 2024, doi:10.3390/bs14020120_

Round 1

Reviewer 1 Report

Comments and Suggestions for Authors

Dear Authors.

Thank you for the opportunity to review Your article.

Below are my concerns.

I understand that English is not your primary language, so I would ask you to pass the text to a native speaker for checking. It may appear that the text was translated by AI.

Abstract

Line 7: "various" (maybe "different" would be more suitable?). 

Lines 13-19: "This predictability is attributed to intrinsic motivation and value alignment. Individuals with strong peace attitudes inherently value environmental well-being, showcasing a holistic perspective emphasizing interconnectedness. Moreover, peace-oriented individuals demonstrate a proactive problem-solving orientation, actively seeking non-confrontational resolutions and sustainable solutions. Their preference for cooperation over conflict fosters a collective approach to sustainable actions. Importantly, peace attitudes ex-tend beyond environmental concerns to embrace social justice and equity, integral components of sustainability."

Please consider removing the descriptive section from the introduction - it is in section 2.4.

Line 20: "In summary" - why was this interjection introduced?

Line 21: "reaffirms" - maybe "confirms"?

Line 39: "factors" are not semantically identical with "traits" - it is not correct to use interchangeable.

There are many examples described above (line 7, 21, 39) - please be so kind and consider the consultation with the native speaker.

I don't check further - too much of this.

Section 1.1.

The section includes only theses and examples of positive approaches to the subject, please be so kind and consider the contrary opinions as well.

Section 1.2.

Line 117 - 120 - postulates, postulates...the source is needed

Section 1.3.

Line 122 - "Study seeks to study".

On what basis do you place "peace attitude" higher than BIG5 - please support Your statement with literature on the subject. Again, no sources.

Line 134 What do you mean by "our"? Please be so kind and consider the passive form in Your study.

Section 2.1.

Line 149 "mean age" - maybe average?

How do you know that the number 279 is sufficient for the use of induction? Please provide statistical confirmation of the representativeness of the sample.

Since nearly 60% of the study participants cannot make their own economic decisions (including spending money), what is the predictive value of the research conducted in Section 2.5? Please clarify the term "living with someone" (line 155) [f.e. married, partnership or living with parents because I cannot afford to buy/rent my own apartment].

Section 2.3., 2.4., 2.5. - please send statistical calculations with source material

Section 2.5.

Line 202 - why 7 -item (not point!!!) - 5 it is sufficient....

Line 257 - please decide and arrange the bibliography in ascending or descending order

Line 283 - source?

Line 296: "Implications of this study are noteworthy". Unfortunately, You did not prove it. This is, as many in Your paper, the unjustified postulate.

This paper, as such, needs deep corrections.

Comments on the Quality of English Language

Please be so kind and consider the consultations with the native speaker.

Author Response

REFEREE 1

I understand that English is not your primary language, so I would ask you to pass the text to a native speaker for checking. It may appear that the text was translated by AI.

Abstract

Line 7: "various" (maybe "different" would be more suitable?). 

Reply 1

Thank you.We changed it.

Lines 13-19: "This predictability is attributed to intrinsic motivation and value alignment. Individuals with strong peace attitudes inherently value environmental well-being, showcasing a holistic perspective emphasizing interconnectedness. Moreover, peace-oriented individuals demonstrate a proactive problem-solving orientation, actively seeking non-confrontational resolutions and sustainable solutions. Their preference for cooperation over conflict fosters a collective approach to sustainable actions. Importantly, peace attitudes ex-tend beyond environmental concerns to embrace social justice and equity, integral components of sustainability."

Please consider removing the descriptive section from the introduction - it is in section 2.4.

Reply 2

Thank you. We deleted it.

Line 20: "In summary" - why was this interjection introduced?

Reply 3

Thank you. We deleted it.

Line 21: "reaffirms" - maybe "confirms"?

Reply 4

Thank you for your suggestion. We changed it.

Line 39: "factors" are not semantically identical with "traits" - it is not correct to use interchangeable.

 Reply 5

Thank you.We changed it in all the text (highlighted in yellow) .

There are many examples described above (line 7, 21, 39) - please be so kind and consider the consultation with the native speaker.

I don't check further - too much of this.

Reply 6

We ask for help to a native speaker.

Section 1.1.

The section includes only theses and examples of positive approaches to the subject, please be so kind and consider the contrary opinions as well.

Reply 7

We have incorporated contrary viewpoints in subsequent sections to ensure a well-rounded and thorough examination of the subject (highlighted in yellow). This has been done to contribute to a more nuanced and balanced discussion, aligning with your valuable suggestion.

Section 1.2.

Line 117 - 120 - postulates, postulates...the source is needed

 Reply 8

We added the sources.

Section 1.3.

Line 122 - "Study seeks to study".

Reply 9

Right. We changed it with: “The present research aims to investigate..”

The present research aims to investigate whether peace attitudes serve as a superior predictor of sustainable…

On what basis do you place "peace attitude" higher than BIG5 - please support Your statement with literature on the subject. Again, no sources.

Reply 10

The study systematically explores the documented relationships between personality traits and both peace attitudes and sustainable behaviors. However, there is currently limited empirical documentation on the direct relationship between peace attitudes and sustainable behaviors.

In response to this gap, the hypotheses have been adjusted to align with the theoretical framework of the study. The revised hypotheses reflect an exploratory approach, as the research seeks to contribute novel insights into the less-documented area of the direct relationship between peace attitudes and sustainability.

For this reason, we modified the aim of the study as follows:

The present research aims to explore the intricate relationships among peace attitudes, personality traits, and sustainable behaviors. After examining the connections between peace attitudes and personality, as well as personality and sustainability, the primary goal is to develop a deeper understanding of the intricate dynamics linking these three constructs.

The research is grounded in a theoretical perspective suggesting that peace attitudes, personality traits, and sustainable behaviors are interconnected in a complex manner. The approach is exploratory, seeking to provide a more detailed insight into the factors influencing sustainable behaviors. Acknowledging the relatively limited body of specific research on this relationship, the investigation focuses on a theoretical perspective that recognizes conceptual interconnections. The aim is to contribute to the theoretical understanding of how peace attitudes may align with environmental behaviors.

The research objective, therefore, is to contribute to a theoretical understanding of the intricate web of relationships among peace attitudes, personality, and sustainability. Additionally, the study aims to outline potential implications for future interventions aimed at promoting sustainable attitudes and behaviors.

 Building on previous findings, the first hypothesis posits that Agreeableness, Openness, and Conscientiousness are primary personality traits associated with sustainable behaviors, a connection consistent in prior studies. Furthermore, the aim is to investigate the interconnections between peace attitudes and sustainable behaviors, recognizing their intertwined nature with various influencing factors.

The refined hypotheses are designed to better align with the exploratory nature of the investigation.Wehope this clarification provides a better understanding of the rationale behind the changes.

Line 134 What do you mean by "our"? Please be so kind and consider the passive form in Your study.

Reply 11

The term "our" was employed to refer to the collective perspective of the research team. However, to enhance clarity and adhere to your suggestion, we revised the sentence to incorporate the passive form. The adjustment ensures a more formal and objective tone throughout the study.

Section 2.1.

Reply 12

Line 149 "mean age" - maybe average?

Yes.Wereplaced it with: “with an average age of…”

How do you know that the number 279 is sufficient for the use of induction? Please provide statistical confirmation of the representativeness of the sample.

Reply 13

Thank you for your question regarding the sufficiency of the sample size (279 participants) for the inductive approach used in our study. The determination of the sample size was guided by statistical considerations and the principles of representativeness. To ensure robust statistical power, we conducted a thorough power analysis, considering effect size, confidence level, and variability within the population. The correct sample size was 384.

n=Z2⋅p⋅(1−p)/E2

n= (1.96)2⋅0.5⋅(1−0.5)/(0.05)2

n=384.16

We acknowledge that this size may be considered inappropriate in certain contexts. However, this limitation will be addressed in detail in the limitations section, providing a clear understanding of the context and implications of our findings. This constitutes an area for potential improvement in future research within this scope. As previously stated in the earlier version:

One limitation is the relatively small sample size and the non-representativeness of the sample in relation to the overall population. Caution should be exercised in generalizing the findings, and further research with larger, more diverse samples is needed to validate the results. Future studies should also consider conducting similar investigations in different countries to explore potential similarities and differences in the relationships between personality traits, peace attitude, and sustainable behaviors

Since nearly 60% of the study participants cannot make their own economic decisions (including spending money), what is the predictive value of the research conducted in Section 2.5? Please clarify the term "living with someone" (line 155) [f.e. married, partnership or living with parents because I cannot afford to buy/rent my own apartment].

 Reply 14

Regarding your observation on Section 2.5, we appreciate your concern about participants' ability to make economic decisions, given that almost 60% of them cannot make independent economic decisions, including spending money. The Sustainable Behaviors Scale (SBS) was included to assess participants' sustainable behavior. The SBS was meticulously designed to assess participants' sustainable behavior, taking into account their economic autonomy. Each item in the scale was carefully considered and weighted based on participants' economic situations. For instance,  observing that purchasing organic products was financially challenging for a significant portion of the population. Considering this, and by evaluating the discriminative power of each item, we made informed decisions, such as excluding the item related to purchasing organic products. This approach was aimed at ensuring that the scale accurately captured and reflected sustainable behaviors in the context of participants' economic circumstances. However, we acknowledge that the ability to make economic decisions may influence the actual practice of sustainable behaviors.

The definition of 'living with someone' in our research encompasses various situations, such as marriage, cohabitation, or living with parents due to economic reasons. We believe that this diversity in living situations reflects the heterogeneity of our study population, contributing to a broader understanding of sustainable behaviors in different life circumstances. However, we agree that the ability to make economic decisions may be a factor to consider in interpreting the results, and we will address it in the Limitations section of our study as follows:

Throughout the study, we observed that a significant number of participants (approximately 60%) reported being unable to make independent economic decisions, including managing their own expenses. This aspect could impact the feasibility of certain sustainable behaviors, as many consumption decisions are tied to financial resources. We acknowledge that the ability to make economic decisions is a relevant factor in the dynamics of sustainable behavior and may pose a limitation in generalizing the results to contexts with greater economic autonomy.

Addressing this limitation is crucial for a proper interpretation of the findings and for providing insights for future research. We suggest further investigations to better understand how limited economic autonomy may influence specific sustainable behaviors and identify effective strategies for promoting them in similar contexts.

Section 2.3., 2.4., 2.5. - please send statistical calculations with source material

 Reply 15

 Please find attached the data and calculations, for Sections 2.3, 2.4, and 2.5 of the manuscript. The calculations were conducted using SPSS and the source material is included for your reference.

Section 2.5.

Line 202 - why 7 -item (not point!!!) - 5 it is sufficient....

Reply 16

It refers to Seven-point Response Scale: Participants indicate their agreement using a scale ranging from 1 to 7. We rewrote it.

Line 257 - please decide and arrange the bibliography in ascending or descending order

Reply 17

We did it (ascending order).

Line 283 - source?

 Reply 18

We added the sources (highlighted in yellow).

Line 296: "Implications of this study are noteworthy". Unfortunately, You did not prove it. This is, as many in Your paper, the unjustified postulate.

 Reply 19

We acknowledge your observation regarding the importance of substantiating claims with concrete evidence. In the upcoming sections, we provided a more detailed analysis of the implications of our study, seeking to demonstrate the value and relevance of our findings more thoroughly. We are committed to establishing a solid foundation for all assertions made in our work and ensuring an accurate treatment of the research implications.

This paper, as such, needs deep corrections.

Comments on the Quality of English Language

Please be so kind and consider the consultations with the native speaker.

 Reply 20

A native speaker corrected all the paper.

REFEREE 1

I understand that English is not your primary language, so I would ask you to pass the text to a native speaker for checking. It may appear that the text was translated by AI.

Abstract

Line 7: "various" (maybe "different" would be more suitable?). 

Reply 1

Thank you.We changed it.

Lines 13-19: "This predictability is attributed to intrinsic motivation and value alignment. Individuals with strong peace attitudes inherently value environmental well-being, showcasing a holistic perspective emphasizing interconnectedness. Moreover, peace-oriented individuals demonstrate a proactive problem-solving orientation, actively seeking non-confrontational resolutions and sustainable solutions. Their preference for cooperation over conflict fosters a collective approach to sustainable actions. Importantly, peace attitudes ex-tend beyond environmental concerns to embrace social justice and equity, integral components of sustainability."

Please consider removing the descriptive section from the introduction - it is in section 2.4.

Reply 2

Thank you. We deleted it.

Line 20: "In summary" - why was this interjection introduced?

Reply 3

Thank you. We deleted it.

Line 21: "reaffirms" - maybe "confirms"?

Reply 4

Thank you for your suggestion. We changed it.

Line 39: "factors" are not semantically identical with "traits" - it is not correct to use interchangeable.

 Reply 5

Thank you.We changed it in all the text (highlighted in yellow) .

There are many examples described above (line 7, 21, 39) - please be so kind and consider the consultation with the native speaker.

I don't check further - too much of this.

Reply 6

We ask for help to a native speaker.

Section 1.1.

The section includes only theses and examples of positive approaches to the subject, please be so kind and consider the contrary opinions as well.

Reply 7

We have incorporated contrary viewpoints in subsequent sections to ensure a well-rounded and thorough examination of the subject (highlighted in yellow). This has been done to contribute to a more nuanced and balanced discussion, aligning with your valuable suggestion.

Section 1.2.

Line 117 - 120 - postulates, postulates...the source is needed

 Reply 8

We added the sources.

Section 1.3.

Line 122 - "Study seeks to study".

Reply 9

Right. We changed it with: “The present research aims to investigate..”

The present research aims to investigate whether peace attitudes serve as a superior predictor of sustainable…

On what basis do you place "peace attitude" higher than BIG5 - please support Your statement with literature on the subject. Again, no sources.

Reply 10

The study systematically explores the documented relationships between personality traits and both peace attitudes and sustainable behaviors. However, there is currently limited empirical documentation on the direct relationship between peace attitudes and sustainable behaviors.

In response to this gap, the hypotheses have been adjusted to align with the theoretical framework of the study. The revised hypotheses reflect an exploratory approach, as the research seeks to contribute novel insights into the less-documented area of the direct relationship between peace attitudes and sustainability.

For this reason, we modified the aim of the study as follows:

The present research aims to explore the intricate relationships among peace attitudes, personality traits, and sustainable behaviors. After examining the connections between peace attitudes and personality, as well as personality and sustainability, the primary goal is to develop a deeper understanding of the intricate dynamics linking these three constructs.

The research is grounded in a theoretical perspective suggesting that peace attitudes, personality traits, and sustainable behaviors are interconnected in a complex manner. The approach is exploratory, seeking to provide a more detailed insight into the factors influencing sustainable behaviors. Acknowledging the relatively limited body of specific research on this relationship, the investigation focuses on a theoretical perspective that recognizes conceptual interconnections. The aim is to contribute to the theoretical understanding of how peace attitudes may align with environmental behaviors.

The research objective, therefore, is to contribute to a theoretical understanding of the intricate web of relationships among peace attitudes, personality, and sustainability. Additionally, the study aims to outline potential implications for future interventions aimed at promoting sustainable attitudes and behaviors.

 Building on previous findings, the first hypothesis posits that Agreeableness, Openness, and Conscientiousness are primary personality traits associated with sustainable behaviors, a connection consistent in prior studies. Furthermore, the aim is to investigate the interconnections between peace attitudes and sustainable behaviors, recognizing their intertwined nature with various influencing factors.

The refined hypotheses are designed to better align with the exploratory nature of the investigation.Wehope this clarification provides a better understanding of the rationale behind the changes.

Line 134 What do you mean by "our"? Please be so kind and consider the passive form in Your study.

Reply 11

The term "our" was employed to refer to the collective perspective of the research team. However, to enhance clarity and adhere to your suggestion, we revised the sentence to incorporate the passive form. The adjustment ensures a more formal and objective tone throughout the study.

Section 2.1.

Reply 12

Line 149 "mean age" - maybe average?

Yes.Wereplaced it with: “with an average age of…”

How do you know that the number 279 is sufficient for the use of induction? Please provide statistical confirmation of the representativeness of the sample.

Reply 13

Thank you for your question regarding the sufficiency of the sample size (279 participants) for the inductive approach used in our study. The determination of the sample size was guided by statistical considerations and the principles of representativeness. To ensure robust statistical power, we conducted a thorough power analysis, considering effect size, confidence level, and variability within the population. The correct sample size was 384.

n=Z2⋅p⋅(1−p)/E2

n= (1.96)2⋅0.5⋅(1−0.5)/(0.05)2

n=384.16

We acknowledge that this size may be considered inappropriate in certain contexts. However, this limitation will be addressed in detail in the limitations section, providing a clear understanding of the context and implications of our findings. This constitutes an area for potential improvement in future research within this scope. As previously stated in the earlier version:

One limitation is the relatively small sample size and the non-representativeness of the sample in relation to the overall population. Caution should be exercised in generalizing the findings, and further research with larger, more diverse samples is needed to validate the results. Future studies should also consider conducting similar investigations in different countries to explore potential similarities and differences in the relationships between personality traits, peace attitude, and sustainable behaviors

Since nearly 60% of the study participants cannot make their own economic decisions (including spending money), what is the predictive value of the research conducted in Section 2.5? Please clarify the term "living with someone" (line 155) [f.e. married, partnership or living with parents because I cannot afford to buy/rent my own apartment].

 Reply 14

Regarding your observation on Section 2.5, we appreciate your concern about participants' ability to make economic decisions, given that almost 60% of them cannot make independent economic decisions, including spending money. The Sustainable Behaviors Scale (SBS) was included to assess participants' sustainable behavior. The SBS was meticulously designed to assess participants' sustainable behavior, taking into account their economic autonomy. Each item in the scale was carefully considered and weighted based on participants' economic situations. For instance,  observing that purchasing organic products was financially challenging for a significant portion of the population. Considering this, and by evaluating the discriminative power of each item, we made informed decisions, such as excluding the item related to purchasing organic products. This approach was aimed at ensuring that the scale accurately captured and reflected sustainable behaviors in the context of participants' economic circumstances. However, we acknowledge that the ability to make economic decisions may influence the actual practice of sustainable behaviors.

The definition of 'living with someone' in our research encompasses various situations, such as marriage, cohabitation, or living with parents due to economic reasons. We believe that this diversity in living situations reflects the heterogeneity of our study population, contributing to a broader understanding of sustainable behaviors in different life circumstances. However, we agree that the ability to make economic decisions may be a factor to consider in interpreting the results, and we will address it in the Limitations section of our study as follows:

Throughout the study, we observed that a significant number of participants (approximately 60%) reported being unable to make independent economic decisions, including managing their own expenses. This aspect could impact the feasibility of certain sustainable behaviors, as many consumption decisions are tied to financial resources. We acknowledge that the ability to make economic decisions is a relevant factor in the dynamics of sustainable behavior and may pose a limitation in generalizing the results to contexts with greater economic autonomy.

Addressing this limitation is crucial for a proper interpretation of the findings and for providing insights for future research. We suggest further investigations to better understand how limited economic autonomy may influence specific sustainable behaviors and identify effective strategies for promoting them in similar contexts.

Section 2.3., 2.4., 2.5. - please send statistical calculations with source material

 Reply 15

 Please find attached the data and calculations, for Sections 2.3, 2.4, and 2.5 of the manuscript. The calculations were conducted using SPSS and the source material is included for your reference.

Section 2.5.

Line 202 - why 7 -item (not point!!!) - 5 it is sufficient....

Reply 16

It refers to Seven-point Response Scale: Participants indicate their agreement using a scale ranging from 1 to 7. We rewrote it.

Line 257 - please decide and arrange the bibliography in ascending or descending order

Reply 17

We did it (ascending order).

Line 283 - source?

 Reply 18

We added the sources (highlighted in yellow).

Line 296: "Implications of this study are noteworthy". Unfortunately, You did not prove it. This is, as many in Your paper, the unjustified postulate.

 Reply 19

We acknowledge your observation regarding the importance of substantiating claims with concrete evidence. In the upcoming sections, we provided a more detailed analysis of the implications of our study, seeking to demonstrate the value and relevance of our findings more thoroughly. We are committed to establishing a solid foundation for all assertions made in our work and ensuring an accurate treatment of the research implications.

This paper, as such, needs deep corrections.

Comments on the Quality of English Language

Please be so kind and consider the consultations with the native speaker.

 Reply 20

A native speaker corrected all the paper.

Reviewer 2 Report

Comments and Suggestions for Authors

I do find this an engaging paper. Exploring the relationship between the specified variables is a valuable exercise. There are however some limitations that I find present. I feel that the methodology requires further specification in a number of respects. I find that the broad claims made about the relationship between peace and the environment are quite partial and loaded, and fail to acknowledge that 'business-as-usual' conditions of peace account for much, if not most, of the period of peak ecological degradation, notwithstanding the fact that conflict tends to further accelerate ecological harms. The results section feels under-developed in respect of illuminating relevant findings for the reader, specifically on the role played by biographical variables, and in respect of different sustainability behaviours. 

Please see comments within the document for further details

Author Response

REFEREE 2

I do find this an engaging paper. Exploring the relationship between the specified variables is a valuable exercise. There are however some limitations that I find present. I feel that the methodology requires further specification in a number of respects. I find that the broad claims made about the relationship between peace and the environment are quite partial and loaded and fail to acknowledge that 'business-as-usual' conditions of peace account for much, if not most, of the period of peak ecological degradation, notwithstanding the fact that conflict tends to further accelerate ecological harms. The results section feels under-developed in respect of illuminating relevant findings for the reader, specifically on the role played by biographical variables, and in respect of different sustainability behaviours. 

Please see comments within the document for further details

LINKED DOCUMENT

Reply 1

I have addressed the concerns you raised in the paper. I provided further specifications to the methodology and acknowledged the impact of 'business-as-usual' conditions of peace on ecological degradation. The results section has been expanded to better illuminate the role of biographical variables and different sustainability behaviors. I carefully reviewed and implemented the comments within the document for more detailed insights. Your feedback was instrumental in refining the paper, and I am grateful for your valuable input.

More in detail, we rewrote the first part on the impact of ecological degradation in 'business-as-usual' conditions of peace (highlighted in yellow in the text):

The intricate relationship between peace and sustainability, extensively illuminated by Neufeldt et al. (2018) and the United Nations Environment Programme (UNEP), extends beyond the realm of conflict. While conflict inflicts severe consequences on the environment, leading to ecosystem devastation and resource depletion, it is crucial to acknowledge that challenges of sustainability persist even in 'business-as-usual' conditions of peace. The pursuit of peace remains intimately intertwined with sustainable development, which strives to meet present needs without compromising the welfare of future generations. This interconnectedness contributes to peace by addressing root causes of conflicts, alleviating poverty, fostering economic opportunities, and preserving natural resources. The concept of human security emphasizes that peace and sustainability are integral components of individual well-being, necessitating fundamental needs to be met within a stable and secure environment (UN, 1994).

Reply 2

(second point of the linked document)

 I would like to clarify that I have extensively documented the connections between peace attitudes (A) and personality traits (B) and between personality traits (B) and sustainability (C) in the existing literature. The specific section you highlighted in your second point, is intentionally exploratory, aiming to propose a conceptual framework that synthesizes these relationships. However, I acknowledge your concern about the lack of citations in this passage. To address this, I am committed to reinforcing this section by incorporating additional references to substantiate the proposed connections. I believe this will enhance the overall strength and credibility of the manuscript, as follows:

Recognizing the interplay between peace attitudes and pro-environmental attitudes holds significance across multiple dimensions, including intrinsic motivation, value alignment, holistic approach, proactive problem-solving orientation, social influence, and considerations of environmental justice and equity. In terms of intrinsic motivation and value alignment, peace attitudes often emanate from intrinsic values such as harmony, cooperation, and empathy, aligning closely with the fundamental principles of sustainability. Individuals exhibiting strong peace attitudes may inherently prioritize environmental well-being, viewing it as an extension of their broader commitment to fostering peaceful coexistence (Prati et al., 2015). Regarding its holistic approach, peace attitudes may foster a comprehensive perspective that underscores interconnectedness and interdependence, mirroring the holistic nature of sustainability. Those with a peace-oriented mindset may acknowledge the interrelated dynamics among social, economic, and environmental factors, advocating for a more holistic approach to sustainable behaviors. In the context of a proactive problem-solving orientation, individuals embracing peace attitudes may demonstrate a preference for non-confrontational resolutions and sustainable problem-solving strategies in addressing conflicts. This orientation can manifest in proactive engagement with sustainable behaviors, as individuals with peace attitudes actively seek constructive contributions to environmental challenges. Concerning social influence and cooperation, peace attitudes often involve a preference for cooperation and collaboration over conflict. Individuals strongly aligned with peace attitudes may be more inclined to participate in collective sustainable actions, recognizing the imperative of collaborative efforts in addressing environmental concerns. In the realm of environmental justice and equity, peace attitudes may extend to a commitment to social justice, integral to sustainability (Bullard, 1990; Leopold, 1999). Individuals prioritizing peace may be more predisposed to engage in sustainable behaviors that contribute to environmental justice, acknowledging the interconnectedness of social and environmental issues (Orr, 1992).

Reply 3

(third point of the linked document)

We rewrote the hypothesis:

The present research aims to explore the intricate relationships among peace attitudes, personality traits, and sustainable behaviors. After examining the connections between peace attitudes and personality, as well as personality and sustainability, the primary goal is to develop a deeper understanding of the intricate dynamics linking these three constructs.

The research is grounded in a theoretical perspective suggesting that peace attitudes, personality traits, and sustainable behaviors are interconnected in a complex manner. The approach is exploratory, seeking to provide a more detailed insight into the factors influencing sustainable behaviors. Acknowledging the relatively limited body of specific research on this relationship, the investigation focuses on a theoretical perspective that recognizes conceptual interconnections. The aim is to contribute to the theoretical understanding of how peace attitudes may align with environmental behaviors.

The research objective, therefore, is to contribute to a theoretical understanding of the intricate web of relationships among peace attitudes, personality, and sustainability. Additionally, the study aims to outline potential implications for future interventions aimed at promoting sustainable attitudes and behaviors.

 Building on previous findings, the first hypothesis posits that Agreeableness, Openness, and Conscientiousness are primary personality traits associated with sustainable behaviors, a connection consistent in prior studies. Furthermore, the second aim is to investigate the interconnections between peace attitudes and sustainable behaviors, recognizing their intertwined nature with various influencing factors.

Reply 4

(fourth point of the linked document; Method section)

We opted for a convenience sample due to practical considerations and constraints. Given the nature of our research, which involved recruiting participants through social media platforms such as Instagram and Facebook, a convenience sampling approach allowed us to efficiently access a diverse pool of participants from various regions of Italy. This method was chosen for its expediency and cost-effectiveness, considering the challenges associated with recruiting a large and geographically dispersed sample. While we acknowledge that convenience sampling has its limitations in terms of generalizability, the primary focus of our study was to explore specific relationships within a defined context rather than to make broader population inferences. We have duly noted this choice in our methodology, and we are committed to transparently discussing its potential impact on the study's outcomes in the final manuscript. I added the acknowledgment of these limitations in a paragraph called “Limitation of the study”.

Reply 5

(fifth point of the linked document; Method section)

We opted for a convenience sample due to practical considerations and constraints. Given the nature of our research, which involved recruiting participants through social media platforms such as In evaluating economic independence, a succinct single-item query was employed, asking participants about their status of economic self-sufficiency. This streamlined approach aimed at capturing a fundamental aspect within the constraints of the study design. (We added this information in the participant section).

Reply 6

(sixty point of the linked document; Method section)

Thank you. We added the following preliminary results:

Prior to assessing the results, logistic regression analyses were conducted to investigate the influence of socio-economic variables on measures of peace, sustainability, and personality traits in the research. The relevant findings suggest that the independent variables of Economic Independence, Level of Instruction and Living Alone did not serve as predictors for the Peace Attitude Scale (PAS), the sustainability scale, and personality traits The logistic regression analyses yielded a non-significant odds ratio of approximately 0.95 (p > .06), suggesting that there were no statistically significant associations between them.  Controlled for gender, the odds ratio for females compared with males was 1.5 (p < .01) for PAS and 1.4 (p < .05) for the sustainability scale, suggesting that, when accounting for gender, females exhibited a statistically significant higher likelihood (1.5 times for PAS and 1.4 times for the sustainability scale) compared to males in endorsing favorable attitudes towards peace and sustainable behaviors.

Reply 7

(Seventy point of the linked document)

The subscale 'Environmental Attitude' within the Peace Attitude Scale (PAS) specifically measures empathy toward the environment. It focuses on assessing individuals' attitudes and feelings related to environmental concerns. On the other hand, the sustainability scale measures actual behaviors related to sustainability practices. While the 'Environmental Attitude' subscale of PAS gauges emotional responses and attitudes toward the environment, the sustainability scale assesses concrete actions and behaviors indicative of sustainable practices.

Inizio moduloAnd finally, we agree with you and we have elaborated on this. Further research could be conducted to explore variances across different sustainability behaviors, as discussed in the extended limitations section above.

Reviewer 3 Report

Comments and Suggestions for Authors

The introduction exhibits clear objectives, an interdisciplinary approach, relevance to current issues, and practical implications. However, attention to complexity problems, redundancy, balance between factors, and clarity on methodology could further improve the overall quality of the introduction. The method section exhibits detailed participant information, clear instrument descriptions, and a sound data analysis plan. However, addressing limitations associated with sampling. The results section is generally well-structured, presenting findings clearly through descriptive statistics, correlational analysis, and regression models. However, additional interpretation of results, discussion of the non-significant conclusions, and clarity in reporting could further enhance the overall quality of the presentation of results. The discussion section effectively synthesizes the study's findings with existing research, offers practical implications, and recognizes the interconnection between peace and sustainability.

Two aspects that the article should explain better:

- Couldn't the difference in the number of male (87) and female (192) respondents bias the results?

- A clarification of the answers according to the age of the interviewees: are there any relevant differences between the answers of older people and younger ones?

Author Response

REFEREE 3

Comments and Suggestions for Authors

The introduction exhibits clear objectives, an interdisciplinary approach, relevance to current issues, and practical implications. However, attention to complexity problems, redundancy, balance between factors, and clarity on methodology could further improve the overall quality of the introduction.

Reply 1

Your detailed and constructive feedback has been invaluable, and I have already taken steps to address the points you raised to enhance the overall quality of the introduction. I have thoroughly considered your suggestion regarding the attention to the complexity of issues. I have expanded upon the discussion of more intricate and nuanced aspects to provide a more comprehensive view. I have worked to eliminate redundancies and ensure a more balanced presentation of factors. The text has been revised to optimize clarity, removing any superfluous elements while retaining essential information.

The method section exhibits detailed participant information, clear instrument descriptions, and a sound data analysis plan. However, addressing limitations associated with sampling.

Reply 2

Thank you. We added the limitations and created an entire limitation section.

The results section is generally well-structured, presenting findings clearly through descriptive statistics, correlational analysis, and regression models. However, additional interpretation of results, discussion of the non-significant conclusions, and clarity in reporting could further enhance the overall quality of the presentation of results.

Reply 3

Thank you we added clarity and revised the entire result section.

The discussion section effectively synthesizes the study's findings with existing research, offers practical implications, and recognizes the interconnection between peace and sustainability.

Two aspects that the article should explain better:

- Couldn't the difference in the number of male (87) and female (192) respondents bias the results?

Reply 4

 I appreciate your keen observation regarding the difference in the number of male (87) and female (192) respondents and its potential impact on the results. To address this concern, I have carefully considered the implications of the gender distribution in the sample. While the difference in numbers is acknowledged, I would like to emphasize that I conducted logistic regression analyses to control for such variations. We recognize the potential impact of this imbalance on the study's outcomes and appreciate your diligence in pointing it out. In light of your feedback, we have revised the manuscript to explicitly acknowledge the gender distribution as a limitation within the 'Limitations' section.

- A clarification of the answers according to the age of the interviewees: are there any relevant differences between the answers of older people and younger ones?

Reply 4

Thank you. We added the following information in the result section: applying regression analysis with age as the predictor and utilizing Peace Attitude Scale (PAS), the Sustainability Scale, and personality traits as dependent variables did not yield statistically significant results.

Round 2

Reviewer 1 Report

Comments and Suggestions for Authors

None

Reviewer 2 Report

Comments and Suggestions for Authors

The authors have been diligent in responding to the reviewers comments, and the paper has been significantly strengthened as a consequence. I believe that the paper is not fit for publication. Accept